# Stabilizing Total Mixed Ration Dry Matter to Mitigate Environmental-Relative-Humidity Effects on Lactating Cow Performance

**DOI:** 10.3390/ani15081137

**Published:** 2025-04-15

**Authors:** Yi-Hsuan Chen, Yi-Ming Chen, Po-An Tu, Ching-Yi Chen, Han-Tsung Wang

**Affiliations:** 1Northern Region Branch, Taiwan Livestock Research Institute, Ministry of Agriculture, Sihoo Township, Miaoli 368003, Taiwan; hsuan@mail.tlri.gov.tw (Y.-H.C.); tpa@mail.tlri.gov.tw (P.-A.T.); 2Civil Affairs Office of Xinwu District, Xinwu District, Taoyuan 327008, Taiwan; pipiugozo@gmail.com; 3Department of Animal Science and Technology, National Taiwan University, Da’an District, Taipei 106032, Taiwan; ronichen@ntu.edu.tw

**Keywords:** dry matter, energy-corrected milk, hay, moisture stabilization, precision feeding, temperature–humidity indexes

## Abstract

Environmental relative humidity (ERH) fluctuations can impact the dry matter (DM) content in hay and in a total mixed ration (TMR), potentially affecting the stability of milk production in dairy cows. This study investigated whether precise TMR DM adjustments could mitigate these effects. We first observed that grass hay exhibited greater DM variability than alfalfa hay at a high ERH. Next, we evaluated how TMR DM influenced milk yield and found that high ERH variability (>25%) caused hay moisture fluctuations, affecting TMR DM stability. Adjusting TMR DM contributed to stabilizing the feed intake and reduce variations in milk composition. Finally, in a feeding trial involving 46 lactating cows over three 28-day periods, we calculated the amount of water required to ensure that the final TMR reached 35% DM based on the measured hay moisture content. Precise TMR DM adjustments increased milk yield and reduced fluctuations in milk composition under heat stress. This study confirmed that monitoring hay DM and adjusting TMR DM can help stabilize milk production, ensuring a more consistent nutritional intake for dairy cows, particularly in environments with high ERH variability.

## 1. Introduction

Total mixed ration (TMR) feeding stabilizes dry matter intake (DMI) and enhances milk yield in dairy cattle [1,2]. However, there are often discrepancies between expected and actual TMR, resulting from weighing errors and fluctuations in ingredient moisture content, affecting DM during mixing [3]. TMR composition can also vary over short periods, altering nutrient concentrations in feedstuff [4] and leading to instability in DMI, nutrient balance, and milk production [5,6].

The impact of climate change on the dairy industry has increased significantly [7,8]. Temperature–humidity indexes (THIs), based on average air temperature and relative humidity, are used to evaluate heat stress levels in dairy cows [9], and novel livestock feed is used to reduce the environmental impact [10,11]. Forages constitute the majority of the dry matter in a lactating cow’s TMR. In the diets of high-yield dairy cows, the proportion of forage generally ranges from 45% to 53% on a DM basis. When corn silage is used, the forage content can vary between 41% and 68% on a DM basis; therefore, daily fluctuations in moisture content can significantly impact the composition of nutrients consumed on a given day [12]. Substantial differences in DM content have been reported between hay and corn silage [6]. Changes in TMR DM influence diet palatability and mixing uniformity [13,14], affecting the actual DM and nutrient intake [15]. A consistent DM content improves TMR consistency [15]. During periods of substantial environmental change, it is crucial to monitor the effects of ambient temperature and humidity on hay DM to improve TMR consistency. Furthermore, during periods of heat stress, the DMI of cattle can decrease considerably, leading to greater discrepancies between the intended and actual daily DMI and the nutrient intake of cows. Precision feeding, in which the dietary nutrient supply is adjusted to match the dietary requirements [16], is an effective strategy for improving sustainability and profitability. We hypothesized that changes in environmental relative humidity (ERH) would affect the DM content of hay, thereby altering the DM content of a TMR when hay is added to a TMR mixer. Inconsistencies in TMR composition can negatively affect the production performance of cattle. Therefore, this study investigated the effects of precision feeding on the consistency of the TMR and the production performance of lactating cows by adjusting the TMR DM. We analyzed changes in the moisture content of dried hay under different humidity conditions to confirm the effect of ERH on hay. Subsequently, we analyzed the milk yield of cows that had consumed hay stored under different humidity conditions and its composition to determine whether changes in hay and TMR DM caused by ERH fluctuations affected the performance of lactating cows. Finally, we measured the hay DM content on site to adjust the amount of water added during TMR preparation and ensure consistency in TMR DM.

## 2. Materials and Methods

### 2.1. Ethics

The experiments were conducted in accordance with the Republic of China (Taiwan) regulations and approved by the Institutional Animal Care and Use Committee (IACUC) of the Livestock Research Institute (approval number: 109-9).

### 2.2. Experiment 1: Effects of ERH on Hay DM

In Experiment 1, we collected alfalfa, oat hay, and Timothy hay from a commercial dairy farm in Nantou County, Taiwan, from May to October 2020. The tested hays in May 2020 were prepared from a commercial dairy farm located in central Taiwan (24.002422° N, 120.648911° E). The tested hays were stored in a warehouse at the commercial dairy farm to prevent exposure to sunlight and rain. All the grass hay was sampled twice a month, and each sample was individually collected from 10 bales of hay and then mixed using the grab sampling method. The forage sources included commercially available large square bales of oat hay, Timothy hay, and alfalfa hay. To investigate the effects of ERH on hay DM, we measured the difference in hay DM (%). This difference was calculated by comparing the DM contents of samples exposed to 10% and 90% ERH levels. First, the samples were dried at 65 °C for 2 days in a laboratory oven to reduce their moisture content to less than 10%. Subsequently, we set a humidity chamber to a relative humidity level of 10% or 90% and incubated 250 g of these dry samples at 30 °C for 6 h to evaluate the water absorption capacity of the hay. The measurement method was modified from Chen et al. [17].

### 2.3. Experiment 2: Effects of Changes in TMR DM on Production Performance

#### 2.3.1. Animals and Treatments

Experiment 1 examined how ERH affects hay DM, establishing a relationship between humidity and hay DM. Experiment 2 investigated the effect of variations in ERH on TMR DM, demonstrating how feed consistency impacts production performance. Experiment 2 was conducted from May to September 2020 on a commercial dairy farm in Nantou County, Taiwan, that housed 160 lactating Holstein cows averaging 98 days in milk (DIM) and with a parity of 2.6 ± 1.5. On average, the lactating cows produced 27.8 ± 7.1 kg of milk per day, with a fat content of 3.7% ± 0.3% and a protein content of 3.3% ± 0.3%. During the experimental period, the relative humidity of the stored forage at the commercial dairy farm ranged from 50% to 90%, and the daily temperature ranged from 25 °C to 35 °C. We utilized ERH data collected from the commercial dairy farm. To determine the impact of ERH on TMR DM, we categorized the ERH levels according to their natural variability observed at the dairy farm. Subsequently, we conducted a correlation analysis between the daily fluctuations in relative humidity and TMR DM. The cows were housed together in a free stall barn equipped with rubber beds and solid concrete floors and did not have access to pasture during the study period. The cattle were milked and fed a TMR twice daily (milked at 05:00 and 17:00 and fed at 05:00 and 15:00). The cows were provided free access to fresh water and salt blocks. During the experimental period, the barn was outfitted with a cooling system that included fans and a water spray system.

Due to considerable fluctuations in the hay DM and its substantial contribution to the TMR formulations (approximately 50%), we analyzed the changes in hay DM under different daily ERH levels. We investigated the effect of high and low ERH levels on TMR DM in both high- and low-yield cow groups. We collected 84 TMR samples from the high- and low-yield cows across all treatment groups. On the basis of the assumed and actual intake, we analyzed the changes in DMI and protein intake in the TMR samples from the high- and low-yield cattle groups. The mean DMI levels were calculated based on the actual intake of high- and low-milk-yield dairy cows.

The lactating cows were classified into low-yield (20.6 kg/d) and high-yield (41.7 kg/d) groups based on a milk production threshold of 30 kg/d. Each group received a TMR formulated according to the guidelines of the National Research Council Committee on Animal Nutrition [18]. The ingredient proportions and nutrient composition are detailed in Table 1.

The environmental temperature and relative humidity in the barn were recorded automatically by the Caotun Weather Station of Central Weather Administration (https://www.cwa.gov.tw/V8/C/W/OBS_County.html?ID=10008 (accessed on 31 October 2020)) during the study period.

#### 2.3.2. Effect of Changes in Hay DM on TMR DM and Actual Intake by Cattle

We recorded the ERH and storage temperature of forage at the dairy farm and measured the moisture content of hay before it was used for TMR preparation. During the experiment, the actual intake of high-yield and low-yield dairy cows was calculated seven times every two weeks. The total amount of feed provided to the cows was recorded, while the feed refusals (leftover feed) were collected and weighed at 10:00 and 20:00. The actual DMI was then determined by subtracting the feed refusals from the total feed offered. Differences between the assumed and the actual intake of DM and crude protein in the high- and low-yield cow groups were analyzed. During the experimental period, 14 individual TMR samples were collected from each feeding group each month. To ensure the accuracy and representativeness of the samples, three replicates were collected from the same batch of TMR and analyzed.

#### 2.3.3. Effect of TMR DM on Production Performance at Different Yield Levels

To monitor the effect of the lactation stage on production performance, data on TMR moisture content and milk production were collected over two 7-day intervals each month. During each interval, we analyzed the moisture content of the TMR samples and the milk yield and composition on a daily basis. All milk production data, including the milk fat and protein concentrations, were considered in terms of the ERH levels at which the DM hay that the cattle consumed was stored. The experimental cows were divided into four groups on the basis of their milk yields: <20, 20–30, >30–40, and >40 kg/d. For each interval, the means and standard deviations of milk production and composition were calculated, and the coefficient of variation (CV%) relative to the mean of the TMR DM was plotted. With consideration of the farm’s mating management and available feeding space, we selected fifteen cows with complete milk production and dairy herd improvement (DHI) records from each group to investigate the effect of TMR DM on production performance.

### 2.4. Experiment 3: Effects of Precise TMR DM Adjustment on Production Performance at Different THI Values

#### 2.4.1. Animals and Experimental Design

Changes in ERH influenced the THIs, leading to variations in TMR DM, which ultimately affected milk production in cattle, as demonstrated in Experiments 1 and 2. Since Experiment 3 evaluated whether precise adjustments to TMR DM could mitigate production performance fluctuations under different THI conditions, it played a crucial role in understanding this relationship. Experiment 3 was conducted from August to October 2020 at a dairy farm at the Northern Region Branch of the Livestock Research Institute in Miaoli County, Taiwan to investigate the effect of precise TMR DM adjustment on production performance.

Temperature and relative humidity in the barn were recorded by the Northern Region Branch Weather Station of Central Weather Administration (https://www.cwa.gov.tw/V8/C/W/OBS_County.html?ID=10005 (accessed on 31 October 2020)) during the experimental period. The THI was calculated using the formula THI = (1.8 × T + 32) − [(0.55 − 0.0055 × RH) × (1.8 × T − 26)], where T represents the air temperature (°C), and RH represents relative humidity (%) [19]. A THI value of 72, indicating heat stress in cattle [20], was used as the threshold to assess the THI impact.

The experiment included 46 lactating Holstein cows with an average parity of 2.5 ± 1.6; 18 cows were 0–90 DIM, and 28 cows were 91–150 DIM. All cows were housed together in a free stall equipped with rubber beds and solid concrete floors. The cows did not have access to pasture during the study period. To prevent the influence of the moisture present, such as in silage, on the TMR DM, hay was used as the main ingredient in the TMR formula detailed in Table 1. Moreover, to reduce mechanical errors caused by the mixer wagon, the wagon was calibrated at the beginning of the experiment. The cows were milked twice daily (05:00 and 16:00) and had free access to clean water and salt blocks.

We used a switch-back design to account for natural variations in milk yield due to lactation stage and individual differences [21]. The experimental design consisted of three 28-day periods. During the first two periods, the THIs ranged from 72 to 82, with an average value of approximately 80. In the third period, the THIs ranged from 68 to 76, with an average value of approximately 74. To minimize the effects of extreme temperatures on the experimental outcomes, the dairy barn was equipped with cooling systems, including fans and a water spray system. These measures were implemented based on ambient temperature changes to reduce the interference of excessively high temperatures. The TMR formulation aimed to achieve a theoretical value of 35% DM. During the last two weeks of each 28-day period, we controlled the TMR DM by measuring the moisture content of hay before each TMR preparation. Based on the measured hay’s moisture content, we calculated the amount of water needed to ensure that the final TMR reached 35% DM. In the subsequent two weeks of each period, the TMR was prepared according to the original theoretical formula without measuring the hay moisture content. This approach was repeated three times (a total of three 28-day periods), allowing for statistical comparisons between controlled and uncontrolled periods. During the last 14 days of the experiment, approximately 500 g each of Bermuda grass hay, alfalfa hay, soybean hulls, and wheat bran were sampled for moisture measurement before preparing the TMR. Then, the TMR DM was set to 35% DM, and additional water was calculated and added based on the moisture content of Bermuda grass hay, alfalfa hay, soybean hulls, and wheat bran to achieve a TMR DM content of 35% DM. In addition, this study included two milking periods (0–90 and 91–150 DIM) to investigate the effects of TMR DM adjustment on milk yield and composition.

#### 2.4.2. TMR Sampling and Analysis

Each TMR sample, weighing between 0.8 and 1 kg, was collected within an hour of feeding the cows. To ensure the reliability of the collected data, each batch of TMR was sampled three times, and the mean value was used for the analysis. To prevent contamination with cow saliva, the first layer was removed from the top of the pile before collection. All samples were immediately frozen at −20 °C until further analysis. For DM and chemical analysis, the samples were oven-dried at 65 °C for 48 h, ground, filtered, and analyzed using a 20-mesh sieve. The analysis was conducted according to the methods described by the Association of Official Analytical Chemists [22].

#### 2.4.3. Milk Production and Composition

In Experiments 2 and 3, milk yield was recorded at each milking throughout the experimental period using the DHI system and AFIMEN management system (Afimilk Ltd., Afikim, Israel), respectively. The composition of the milk, including milk fat, protein, and solids-not-fat, was analyzed and recorded using the DHI laboratory system with FOSS MilkoScan FT (FOSS, Hillerød, Denmark). The yield of energy-corrected milk (ECM) was calculated using the milk yield and component concentrations from each milking. ECM was calculated as [(0.327 × kg of milk) + (12.95 × kg of milk fat) + (7.20 × kg of milk protein)]. This equation adjusts the milk output to a 0.68 Mcal/kg energy basis [23]. Differences in milk yield and composition were calculated for each animal during the last 7 days of every experimental period as follows:$$\Sigma \;(X_{n}-X_{n-1})$$

*X*: milk yield or milk composition.

*n*: Sampling days in each treatment section; *n* = 8 to 14.

### 2.5. Statistical Analysis

Data analysis was performed using SAS 9.4 (SAS Institute, Cary, NC, USA). The hay moisture content (Experiment 1) and milk production and composition (Experiment 3) were analyzed using the PROC MIXED procedure for repeated measures. Tukey adjustment for multiple comparisons was applied to control the error rate and reduce the risk of Type I errors when performing pairwise comparisons. The data were transformed for statistical analysis. The associations between ERH and hay moisture content and TMR DM, as well as the associations between TMR DM and milk yield and composition in cows with either high or low milk yields, were analyzed using regression analysis within the MIXED procedure of SAS 9.4 (Experiments 1 and 2). Treatment comparisons were performed using Tukey adjustment for multiple comparisons. *p* values > 0.05 and ≤0.10 were considered to indicate a trend, whereas *p* values < 0.05 were considered to indicate significance.

## 3. Results

### 3.1. Experiment 1: Effects of ERH on Hay DM

Table 2 shows differences in DM% for different types of hay under high (RH% = 90) and low (RH% = 10) ERH levels. The mixed-grass hay (a 1:1 mixture of Timothy hay and oat hay) sorbed more water than the other tested hay samples. In addition, the alfalfa hay demonstrated a relatively stable DM% under high and low ERH levels.

### 3.2. Experiment 2: Effects of Changes in TMR DM on Production Performance

#### 3.2.1. Effect of Hay DM on Changes in TMR DM and Actual Intake by Cattle

Table 3 presents the changes that occurred in the moisture content of the hay used in the TMR. The simple regression analysis results in Figure 1 show a positive association between ERH differences and moisture contents in alfalfa (R^2^ = 0.56, *p* = 0.03) and grass hay (R^2^ = 0.76, *p* = 0.004).

Figure 2 illustrates that when ERH > 75%, no significant association was found between ERH difference and TMR DM in the high-yield (R^2^ = 0.08, *p* = 0.69) and low-yield groups (R^2^ = 0.008, *p* = 0.74). However, when ERH < 75%, a negative association was observed in both high-yield (R^2^ = 0.68, *p* = 0.001) and low-yield (R^2^ = 0.39, *p* = 0.01) groups.

Figure 3 shows that TMR DM decreased significantly in the high-yield (R^2^ = 0.07, *p* = 0.0003) and low-yield groups (R^2^ = 0.66, *p* = 0.0006) when ERH differences exceeded 25%. No significant associations were observed when ERH differences were <25%.

To investigate the effects of TMR DM on DMI and CP intake, we divided the high-and low-yield cattle groups into different TMR DM groups. Table 4 shows the changes in the DMI and protein intake in the TMR samples from the high- and low-yield cattle groups for different TMR DM values.

#### 3.2.2. Effect of TMR DM on Production Performance in Cows with Different Yield Levels

Figure 4 shows a significant negative association between TMR DM (%) and the CV of milk production, fat, protein, and total solids (*p* < 0.05) in the high-yield group (30–40 kg/cow/d). A positive association was observed between TMR DM (%) and milk production (R^2^ = 0.82, *p* < 0.05). For cows yielding >40 kg/cow/d, TMR DM (%) was negatively associated with the CV of milk production and total solids (*p* < 0.05) but not with that of milk protein (*p* > 0.05). A positive association was noted between TMR DM (%) and both milk production and the CV of milk fat (*p* < 0.05).

In the low-yield group (20–30 kg/cow/d), Figure 5 shows significant negative correlations between TMR DM (%) and the CV of milk fat, protein, and total solids (*p* < 0.05), along with a positive correlation between TMR DM (%) and milk production (*p* < 0.05). No significant association was found between milk production and TMR DM (%) in the <20 kg/cow/d group (*p* > 0.05), though a negative trend for total solids was observed (*p* < 0.1).

### 3.3. Experiment 3: Effects of a Precise Formulation of TMR DM on Production Performance Under Different THI Values

Table 5 shows that the TMR DM adjustment significantly increased milk yield in Experiment 2 (*p* = 0.04) when THIs ranged from 65 to 82. However, no significant differences were observed in milk fat (*p* = 0.13) or protein levels (*p* = 0.10) between the adjusted and the non-adjusted groups. When stratified by THI, no significant differences in milk yield, fat, or protein were observed at THI > 72 (*p* > 0.05). The cows receiving adjusted TMR DM had a more stable fat-to-protein ratio (FPR), while the non-adjusted group exhibited greater variability in milk composition.

Table 6 shows the results of Experiment 3, examining the effect of TMR DM adjustment on milk yield and composition and ECM. For cows with DIM < 90, the adjustment group had significantly lower differences in milk yield (*p* = 0.014), ECM (*p* = 0.020), and milk protein percentage (*p* = 0.018) compared to the non-adjustment group. No significant difference was noted in milk fat percentage (*p* = 0.261). For cows with DIM of 90–150, the adjustment group exhibited significantly lower differences in milk yield, milk fat percentage, and ECM (*p* < 0.05), while no significant differences in milk protein percentage were observed (*p* > 0.05).

Figure 6 illustrates that the THIs influenced milk production, but less variation in milk yield occurred during the TMR DM adjustment period (white background) compared to the non-adjustment period (yellow background), with only a 2.5% difference. Greater fluctuations were noted between days 42 and 56 due to the THI changes. This experiment underscores the importance of daily TMR DM stabilization in reducing variations in milk yield and ECM.

## 4. Discussion

### 4.1. Effects of ERH on Hay DM

The results of Experiment 1 revealed that ERH affects the water content in hay, with variations of more than 6% noted in the DM of the mixed grass hay in this study (Table 2). Research on the correlation between ERH and hay DM is limited. Bouasker et al. [24] demonstrated that certain types of straw exhibit a low bulk density and a high water absorption capacity, as indicated by the weight difference between a sample immersed in water and a dry sample at room temperature. Our interpretation of differences in DM is similar to that described by Bouasker et al. [24]; however, we considered the difference in terms of ambient temperature and humidity. These characteristics vary considerably among different varieties of natural fibers, with some exhibiting high moisture sorption [24]. Cellulosic fibers are naturally hydrophilic due to the presence of hydroxyl groups and other polar groups. This hydrophilicity is attributable to the hydroxyl groups of hemicelluloses, cellulose, and lignin [25]. Neutral detergent fiber can be used to estimate the amount of cell wall, including lignin, cellulose, and hemicellulose [26,27]. Generally, leguminous hay has less neutral detergent fiber than grass hay. The neutral detergent fiber of the tested grass hay and leguminous hay in our study averaged 62% and 45%, respectively. Therefore, grass hay and leguminous hay can be presumed to have different water sorption capacities due to variations in their chemical compositions. Furthermore, the water absorption capacity of mixed hay may be affected by a higher number of voids in the mixed stack.

Our findings demonstrated that the moisture content of alfalfa and grass hay increased with the ERH difference within a day (Figure 1). Some studies have indicated that the weather conditions play a crucial role in adding or removing moisture from stored feeds [6,28]; this finding is consistent with that of the present study, which indicated that large fluctuations in ERH, even over a single day, can introduce uncertainty in hay DM prior to its inclusion in a TMR mixer.

### 4.2. Effects of Variations in TMR DM on Production Performance

Rations for dairy cows are typically formulated on the basis of DM, and the actual feed ingredients and total nutrients in the TMR are calculated in accordance with the daily needs of the cattle. The results of Experiment 1 revealed that in lactating cows consuming approximately 50% hay in their TMR, the moisture content in the hay led to a change of at least 2% to 3% in the TMR DM per day. This difference was calculated using half of the 4% to 6% difference in moisture content shown in Table 2 and Table 3. The results of Experiment 2 indicated that the hay used on the experimental dairy farm exhibited a change of approximately 4% to 5% in its moisture content (Table 3). Furthermore, we noted a difference of more than 7% in the DM of the TMR that was prepared on different days. Lactating dairy cattle often consume a diet that differs from the one that they are offered [3], mainly because of feeding errors, such as deviations in weighing, unstable moisture content in the ingredients, and mixing errors during TMR preparation [6]. When the daily ERH was <75% and the ERH difference was >25%, the TMR DM difference in the high-yield group was as high as 5%. Furthermore, we noted a decline in TMR DM as the ERH and ERH differences increased (Figure 2 and Figure 3). Figure 2 shows that ERH < 75% was associated with a greater decline in TMR DM than ERH > 75%. The current results indicate that at low ERH (ERH < 75%), daily humidity fluctuations have a more considerable impact on the TMR DM content, primarily because the hay ingredients absorb water more readily at lower humidity levels. However, under a high ERH, the hay ingredients already contain more moisture, making them less affected by humidity differences. Furthermore, as presented in Figure 2 and Figure 3, TMR DM exhibited fluctuations when the ERH was <75% and the ERH difference was >25% within a day. We noted the changes in TMR DM to be affected more by the daily ERH difference than by the daily average ERH. In addition, changes in TMR DM were higher in the diet of the high-yield cows.

Feeding errors can change the actual composition of the TMR fed to dairy cows, which can affect rumen fermentation stability and nutrient availability [15]. Closer adherence to the theoretical composition of the TMR could help reduce metabolic disorders. We showed that larger fluctuations in ERH, with ERH < 75% and a daily difference > 25%, adversely affected the consistency of TMR DM. These fluctuations likely led to changes in the moisture content of the feed, making it less palatable or harder to consume in consistent amounts. As a result, the cows’ overall feed intake decreased, leading to lower DMI and CP intake (Table 4). This was also reflected in milk production performance. By adjusting the TMR DM, it was possible to overcome the larger fluctuations in ERH that led to a lower DMI, thereby resulting in a higher milk yield (Table 5). In this study, changes in ERH (difference/mean) led to decreases of 4.7% (1.09/22.83) and 4.3% (0.84/19.31) in the DMI of the high- and low-yield cattle, respectively (Table 4). Furthermore, the total daily protein intake declined (difference/mean) by 4.3% (0.15/3.41) and 4.0% (0.11/2.74) in the high- and low-yield groups, respectively (Table 4). When TMR is prepared on-site, the moisture content of hay should be monitored, particularly during periods of considerable ERH fluctuations. Changes in ERH can lead to variations in the DMI of cattle across all production levels, with high-yield cows being more affected. Inconsistency in the nutrient content of the TMR and non-maintenance of the dry-based intake may lead to instability in production performance. Thus, ensuring the stability of nutrient availability in the TMR and managing the effects of ERH on feed components are essential for maintaining the health and productivity of dairy herds.

In this study, long-term monitoring demonstrated a higher TMR DM content to have a positive effect on the stability of dairy cow production and milk composition (Figure 4 and Figure 5). A previous study indicated that TMR DM affects the feeding behavior of dairy cows; the study reported that reducing the DM content from 81% to 64% by adding water could reduce the sorting behavior in lactating cows [14]. Furthermore, adding an appropriate amount of water to the TMR, with DM content within the range of 42–46% and primarily consisting of coarse silage, was reported to reduce the selective feeding behavior and increase DMI and daily milk performance [1,2]. Stabilizing TMR nutrients improved DMI consistency, resulting in higher milk yield and more stable milk fat and protein percentages (Table 5 and Table 6). This indicates that the precise adjustment of TMR DM was beneficial for production performance. The production performance of cattle, particularly high-yield cows, may thus be stable.

### 4.3. Effects of a Precise Formulation of TMR DM on Production Performance Under Different THI Values

The results of Experiment 3, with daily THI values ranging from 65 to 82, indicate that adjusting TMR DM positively affected the production performance during the early and mid-lactation stages (Table 5 and Table 6). Although we observed no significant effect of TMR DM adjustment on milk composition, we noted that differences in milk yield, milk fat, and milk protein decreased with the precision of the TMR DM adjustment (Table 6). During the non-adjusted period, the milk FPR fell below the optimal range of 1.2–1.4 for Holstein cattle [29]. FPR has been reported to be useful as a diagnostic tool for detecting energy deficits [30], subclinical ketosis [31], and subclinical acidosis [32]. Heat stress, which begins at a THI of 72 [20], reduces DMI, rumen digestion, pH, and saliva secretion by decreasing the blood flow to the digestive tract [33]. In this study, the milk FPR during the non-adjusted period was below 1.2 when the THI value was >72, indicating a high risk of acidosis [32]. The FPR value during the adjusted period was 1.22, even when the cows were under heat stress (Table 5). This finding indicates that a precise adjustment of TMR DM is beneficial to milk production and helps maintain consistent DM and nutritional characteristics in TMRs. In addition, it reduces the risk of diseases caused by heat stress, particularly during periods of high THI.

ECM standardizes milk fat and protein variations, enabling consistent production comparisons and accurate energy assessments for diet formulation [34]. The milk yield and composition assay (Table 6) showed that a precise TMR DM adjustment reduced differences in milk yield and ECM during DIM < 90 and 90–150, significantly improving milk production and quality. Milk fat and protein percentage differences were also lower during the adjustment period, indicating improved stability in milk composition.

### 4.4. Practical Implications for Managing TMR DM Under Varying ERH Conditions

DMI is a crucial factor that must be estimated before formulating an animal’s diet [35]. Several studies have investigated the effect of silage moisture on TMR DM and demonstrated that it directly and substantially affects the DM of the final nutrient [4]. In Experiment 3, the diet formulations were specifically designed to include hay, concentrate, and other dry ingredients to eliminate the effect of silage moisture. Therefore, the findings of this study indicate that TMR DM is affected by ERH even without interference from silage moisture. Furthermore, if silage is included in the TMR, the effect of ERH on TMR DM becomes more significant and cannot be neglected.

We investigated the effect of high and low ERH levels on TMR DM in both high- and low-yield cow groups while accounting for potential confounding variables such as feed intake reduction due to environmental stress. Some studies have indicated that reduced feed intake accounts for approximately 35–50% of the total reduction in milk yield during environmentally induced hyperthermia [36]. The mechanistic basis for the reduction in milk yield likely involves multiple systems [37].

Variations in ERH can affect the TMR DM. High-yield cows are more significantly impacted because of their increased feed requirements, making it especially important to pay attention to their needs. Farmers can implement routine hay DM monitoring and TMR adjustments using automated feeding systems to optimize lactating cows’ performance, especially in environments prone to high humidity or heat stress. Moreover, we recommend establishing protocols for managing feed moisture during hot and humid periods to mitigate intake fluctuations.

Seasonal variations, such as changes in ambient temperature and humidity, as well as individual physiological differences among cows, may also influence the results. Further research should explore how different environmental conditions interact with TMR adjustment strategies to develop tailored management practices.

## 5. Conclusions

This study confirmed that fluctuations in ERH (<75% with >25% daily variation) significantly affect hay and TMR DM, particularly in grass hay. Farmers can implement routine hay DM monitoring and TMR adjustments to enhance lactating cow performance, especially in environments prone to high humidity or heat stress. Maintaining TMR DM stabilizes cattle intake and improves milk production. Routine monitoring and precise TMR DM adjustment can enhance production consistency and reduce fluctuations in milk yield, which highlights the potential of this approach as a valuable strategy for dairy farm management. Future research should investigate the long-term economic benefits of consistent TMR DM adjustment and explore strategies for real-time monitoring and automated adjustment in varying climatic conditions.

## Figures and Tables

**Figure 1 animals-15-01137-f001:**
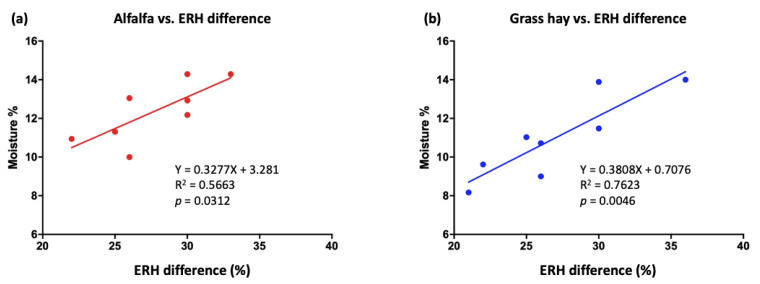
Association between (**a**) alfalfa (red line) and (**b**) grass hay (blue line, 1:1 mixture of oat hay and Timothy hay) moisture contents (%) and environmental relative humidity (ERH) differences (%) within a day (each dot represents the average of 10 bale samples). The hay moisture content was analyzed using the PROC MIXED procedure for repeated measures. The associations between ERH and hay moisture content were analyzed using regression analysis within the MIXED procedure of SAS 9.4. Treatment comparisons were performed using the Tukey adjustment for multiple comparisons. *p* values > 0.05 and ≤0.10 were considered to indicate a trend, whereas *p* values < 0.05 were considered to indicate significance.

**Figure 2 animals-15-01137-f002:**
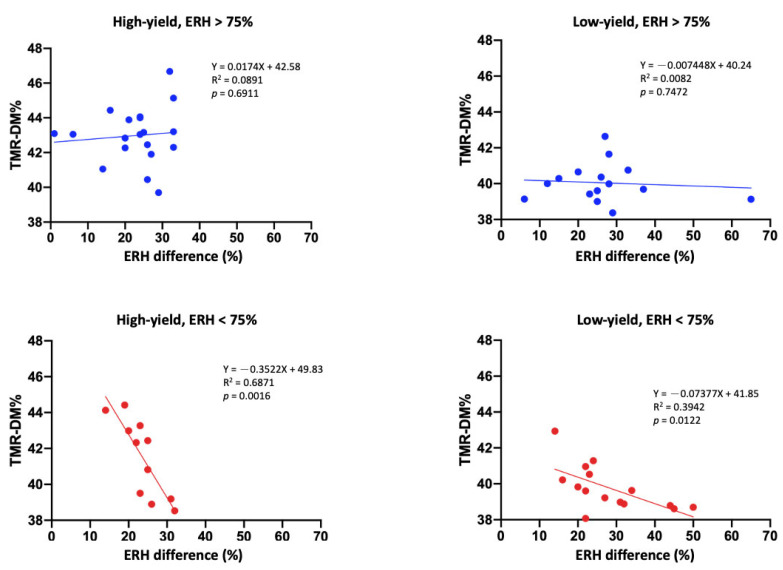
Association between total mixed ration (TMR) dry matter (DM, %) and ERH difference (%) within a day in high- and low-yield groups under environmental relative humidity (ERH) > 75% or <75%. The top figure displays data under ERH > 75% (red lines). The bottom figure displays data under ERH < 75% (blue lines). The data for the high- and low-yield groups were obtained from 84 TMR samples across all treatment groups (each dot represents the average of TMR samples).

**Figure 3 animals-15-01137-f003:**
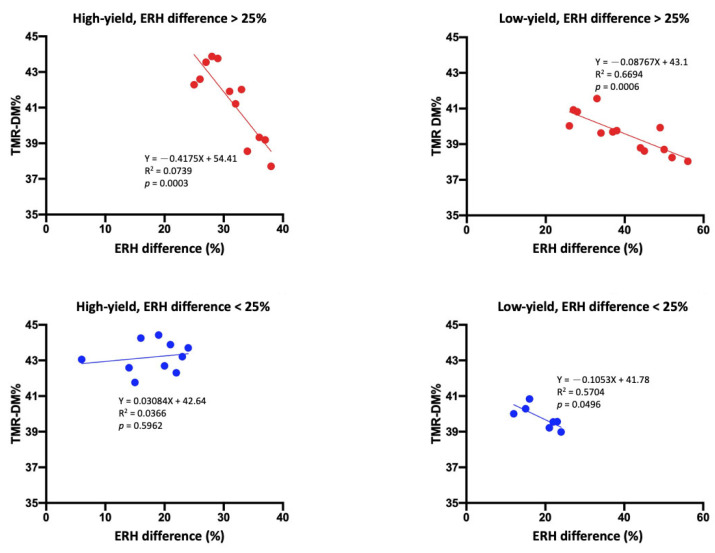
Association between total mixed ration (TMR) dry matter (DM, %) and ERH difference (%) within a day in high- and low-yield groups for environmental relative humidity (ERH) difference > 25% or <25%. The top figure shows the association when the ERH difference was >25% (red lines). The bottom figure shows the association when the ERH difference was <25% (blue lines). The data for the high- and low-yield groups were obtained from 84 TMR samples across all treatment groups (each dot represents the average of TMR samples).

**Figure 4 animals-15-01137-f004:**
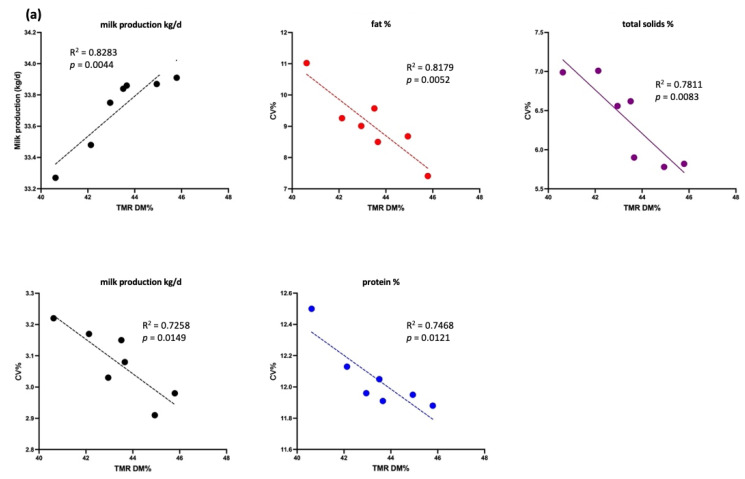
Association between milk production (kg/d) and total mixed ration (TMR) dry matter (DM, %) as well as association between the coefficient of variation (CV, %) of milk production (kg/d, black lines), fat (%, red lines), total solids (%, purple lines), and protein (%, blue lines) and TMR DM (%) in the high-yield group (each dot represents the average milk production and composition in CV% of experimental cows). (**a**) Data for cows producing 30–40 kg/d and (**b**) data for cows producing > 40 kg/d.

**Figure 5 animals-15-01137-f005:**
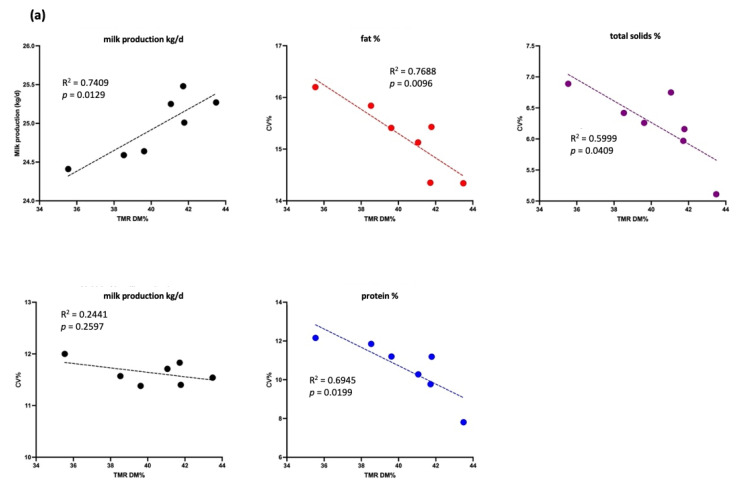
Association between milk production (kg/d) and total mixed ration (TMR) dry matter (DM, %), as well as association between the coefficient of variation (CV, %) of milk production (kg/d, black lines), fat (%, red lines), total solids (%, purple lines), and proteins (%, blue lines) and TMR DM (%) in the low-yield group (each dot represents the average milk production and composition in CV% of experimental cows). (**a**) Data for cows producing 20–30 kg/d and (**b**) data for cows producing < 20 kg/d.

**Figure 6 animals-15-01137-f006:**
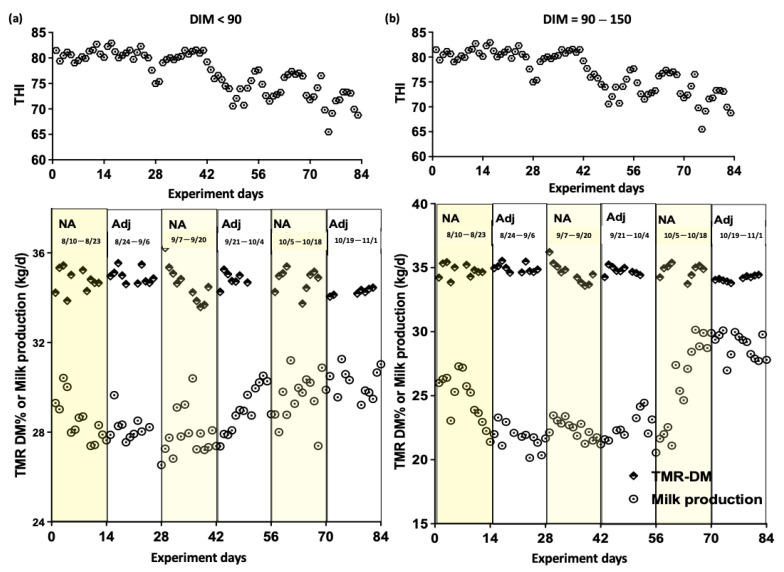
Effect of the temperature–humidity index (THI) and total mixed ration (TMR) dry matter (DM) adjustment on milk production during the experimental period (Experiment 3). (**a**) Cows with days in milk (DIM) less than 90; (**b**) cows with DIM between 90 and 150. NA: non-adjusted; Adj: adjusted.

**Table 1 animals-15-01137-t001:** Ingredients and nutrient composition of total mixed ration (TMR) used as feed in Experiments 2 and 3.

Item	Experiment 2 (% Dry Matter Basis)		Experiment 3 (% Dry Matter Basis)
	High Yield	Low Yield	
Ingredient proportions			
Timothy hay	7.8	14.8	-
Oat hay	19.6	24.6	-
Bermuda grass hay	-	-	29.9
Alfalfa hay	20.6	14.8	20.4
Brewer’s grains, wet	3.4	8.6	-
Soybean hull	3.9	9.9	12.8
Wheat bran	-	-	3.6
Corn	21.5	13.5	16.3
Corn gluten meal	3.1	1.9	2.3
Soybean meal, 44% CP	12.9	8.1	9.8
Fish meal	1.5	1.0	1.2
Molasses	2.2	1.4	1.7
Iodized salt	0.3	0.2	0.3
Sodium bicarbonate	0.9	0.6	0.7
Limestone	0.8	0.5	0.6
Bypass fat	1.0	-	-
Methionine	0.1	0.1	-
Premix ^†^	0.4	0.1	0.4
Total	100	100	100
Nutrient composition			
DM	45.9	43.9	35.9
Crude protein	17.8	16.1	16.0
Crude fat	3.3	2.8	2.3
Neutral detergent fiber	40.9	48.7	45.4
Acid detergent fiber	24.8	29.0	28.1

^†^ Each kilogram of premix contained vit. A, 10,000,000 IU; vit. D3, 1,600,000 IU; vit. E, 70,000 IU; Fe, 50 g; Cu, 10 g; Zn, 40 g; I, 0.5 g; Se, 0.1 g; Co, 0.1 g.

**Table 2 animals-15-01137-t002:** Differences in hay dry matter (DM) under high and low environmental relative humidity (ERH; ERH% = 10% and 90%).

Forage Source ^†^	Difference in DM (%)	DM (%) in ERH (%) = 10 (Maximum)	DM (%) in ERH (%) = 90 (Minimum)
Oat hay	4.88 ± 0.21	88.42 ± 0.22	83.55 ± 0.01
Timothy hay	5.65 ± 1.03	87.81 ± 0.07	82.16 ± 1.10
Mixed-grass hay *	6.15 ± 1.21	88.31 ± 0.59	82.15 ± 1.80
Alfalfa hay	4.60 ± 0.03	89.62 ± 0.11	85.01 ± 0.14

^†^ Each sample was individually collected from 10 bales of hay and then mixed. * 1:1 mixture of Timothy hay and oat hay.

**Table 3 animals-15-01137-t003:** Differences in hay moisture content in Experiment 2 (environmental relative humidity: 50–90%; daily temperature: 25–35 °C).

Source ^†^	Moisture Content (%)	Maximum (%)	Minimum (%)	Difference (%)
Alfalfa hay	12.25 ± 1.51	14.29	10.02	4.29
Grass hay *	10.78 ± 1.93	14.00	8.17	5.83

^†^ Fourteen individual samples for making TMR for each feeding group were collected each month. * Consisting in a 1:1 mixture of oat hay and Timothy hay.

**Table 4 animals-15-01137-t004:** Dry matter intake and crude protein intake of high- and low-yield groups under different TMR DM values in Experiment 2.

Item (kg/cow/d)	High Yield		Low Yield	
Dry matter intake	22.83 ± 0.32		19.31 ± 0.19	
Maximum	23.3		19.67	
Minimum	22.21		18.83	
Difference	1.09		0.84	
Crude protein intake	3.41 ± 0.03		2.74 ± 0.02	
Maximum	3.47		2.79	
Minimum	3.32		2.68	
Difference	0.15		0.11	
	38–41% TMR DM	42–44% TMR DM	38–39% TMR DM	40–41% TMR DM
Dry matter intake	22.15 ± 2.13	23.0 ± 1.64	17.81 ± 2.80	19.10 ± 1.36
Crude protein intake	3.28 ± 0.31	3.43 ± 0.13	2.70 ± 0.41	2.77 ± 0.19

**Table 5 animals-15-01137-t005:** Effect of total mixed ration (TMR) dry matter (DM) adjustment on milk yield and milk composition at temperature–humidity index (THI) levels of 65–82 in Experiment 3.

Item	THI = 65–82				THI > 72				THI ≤ 72			
	NA *	Adj	SEM	*p*-Value	NA	Adj	SEM	*p*-Value	NA	Adj	SEM	*p-*Value
Milk yield (kg/d)	26.987	27.289	0.328	0.041	26.158	26.162	0.861	0.662	28.303	27.786	0.432	0.310
Milk fat (%)	3.716	3.654	0.095	0.131	3.602	3.657	0.271	0.871	3.887	3.653	0.077	<0.001
Milk protein (%)	3.040	2.990	0.056	0.102	3.088	2.985	0.085	0.103	2.965	2.992	0.051	0.378
Milk fat-to-protein ratio	1.222	1.222			1.166	1.225			1.311	1.221		

SEM: standard error of the mean. * NA: non-adjusted; Adj: adjusted.

**Table 6 animals-15-01137-t006:** Effects of TMR DM adjustment on differences in milk yield and milk composition in Experiment 3.

Item	NA ***	Adj	*p*-Value
DIM ^†^ < 90			
Milk yield (kg/d)	1.31 ± 0.34	0.77 ± 0.16	0.014
Fat (%)	0.12 ± 0.03	0.10 ± 0.02	0.261
Protein (%)	0.13 ± 0.02	0.08 ± 0.02	0.018
ECM (kg/d)	0.47 ± 0.06	0.20 ± 0.09	0.020
DIM 90–150			
Milk yield (kg/d)	1.03 ± 0.09	0.66 ± 0.14	0.034
Fat (%)	0.17 ± 0.03	0.11 ± 0.02	0.017
Protein (%)	0.16 ± 0.02	0.12 ± 0.02	0.274
ECM (kg/d)	0.15 ± 0.09	0.09 ± 0.01	0.014

^†^ DIM: days in milk; ECM: energy-corrected milk, calculated as [(0.327 × kg of milk) + (12.95 × kg of milk fat) + (7.20 × kg of milk protein)]. * NA: non-adjusted; Adj: adjusted.

## Data Availability

Data are contained within the article.

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
