# Peer review of "Stabilizing Total Mixed Ration Dry Matter to Mitigate Environmental-Relative-Humidity Effects on Lactating Cow Performance"

_animals, 2025, doi:10.3390/ani15081137_

Round 1
Reviewer 1 Report
Comments and Suggestions for Authors
The manuscript is rich in content. However, the current way of expression makes it difficult for readers to fully understand the study. Here are my concerns:
Title (Lines 2-4): I understand that the author intends to convey two meanings, but using it as a title is confusing. What exactly is the main subject of the study? I suggest the author integrate these two aspects.
Three 28-day periods (Line 22): How did the author handle the effects of the three periods?
Hypothesis in the Abstract (Lines 27-29): It is inappropriate to place the hypothesis in the abstract, especially the first sentence. The abstract should accurately describe the experimental design, which is not addressed in the current version. The author must supplement this.
Abstract (Lines 42-43): It is recommended to add "moisture stabilization" to the abstract and arrange all keywords in alphabetical order.
Animal Management (Lines 233-244): The description of animal management is placed under statistical analysis, which is unreasonable. This also indirectly reflects that the descriptions of experimental design and animal management earlier in the manuscript are not specific enough.
Conclusion (Lines 520-523): It is inappropriate to cite references in the conclusion section. In my view, the content from Lines 527-531 is sufficient as the conclusion.
For the sections of 2. Materials and Methods, 3. Results, and 4. Discussion: Since the manuscript involves three experiments, the expression and transition of results for each experiment are crucial. In the discussion section, the author should integrate these three experiments to better highlight the significance of the study.
Comments on the Quality of English Language
It is suggested that the author polish the language to make it more reader-friendly.
Author Response
Thank you very much for taking the time to review this manuscript. Please find the detailed responses in the attached file and the corresponding revisions and corrections highlighted in red in the re-submitted files.

Reviewer 2 Report
Comments and Suggestions for Authors
The paper, titled "Effect of Environmental Relative Humidity on Total Mixed Ration Dry Matter and Stabilization of the Performance of Lactating Cows Through Total Mixed Ration Moisture Stabilization", addresses an important and timely topic. The study investigates the impact of environmental relative humidity on total mixed ration (TMR) dry matter content and its subsequent effects on dairy cow performance, which is particularly relevant given the challenges posed by climate variability in dairy production. I found the subject matter of the article fascinating and read the manuscript with great interest. The paper aligns well with the scope of the journal, as it contributes valuable insights into precision feeding strategies and their role in optimizing dairy cow nutrition. However, I believe that in its current form, it has several shortcomings that should be addressed to strengthen its scientific rigor and clarity.
Abstract
The abstract clearly outlines the study's objectives and findings; however, it would benefit from more clarity on the practical implications of the results. Consider briefly stating how dairy producers can apply TMR dry matter adjustments in different environmental conditions. Additionally, some statistical results are mentioned, but it would be helpful to specify effect sizes or confidence intervals to provide a clearer sense of the magnitude of the findings.
The phrase “we hypothesized that…” should be revised for clarity, as hypotheses are usually stated in the past tense. Consider rewording it to “This study hypothesized that…” for consistency. The phrase “high ERH variability (>25%)” should be clarified—does this refer to daily fluctuations or seasonal trends? The use of “helped stabilize” in the last sentence is somewhat vague; a more precise term like “contributed to stabilizing” would improve clarity.
Introduction
The introduction effectively highlights the relevance of environmental relative humidity (ERH) and TMR moisture content in dairy production. However, the literature review could be strengthened by including more recent references on the relationship between climate variability and dairy cow nutrition. The hypothesis is clearly stated, but the rationale for choosing specific thresholds for ERH variability (>25%) could be better justified with citations or preliminary data.
The phrase “Forages make up the majority of a lactating TMR” would be clearer if reworded as “Forages constitute the majority of the dry matter in a lactating cow’s TMR.” The citation style is inconsistent in some areas—ensure uniform formatting, particularly for multiple references (e.g., “[7-9]” should use a consistent dash style throughout the manuscript). The term “Precision feeding” should be briefly defined for readers unfamiliar with the concept.
Line 47: I suggest citing 10.1016/j.vas.2025.100434 regarding the DMI and palatability
Line 50 I suggest citing 10.3390/fermentation10080398 and 10.1007/s11250-025-04363-1 regarding variability of feedstuffs.
Line 52 I suggest citing 10.1016/j.vas.2024.100363 regarding heat stress in cattle
Line 60 I suggest citing 10.3389/fvets.2024.1441905 and 10.1080/19440049.2024.2414954 regarding novel feeds
Line 64 I suggest citing 10.3390/ani15030458 regarding precision farming
Materials and Methods
The study design is well-structured, but additional details are needed regarding how TMR samples were collected and analyzed. The frequency of sampling and whether multiple replicates were taken from the same batch should be specified to assess the reliability of the results. The description of the statistical methods is generally clear, but the rationale for using the Tukey adjustment for multiple comparisons should be provided, especially given the types of comparisons being made. If data transformations were applied to meet normality assumptions, these should be explicitly stated.
The acronym “THI” is used without definition in the abstract and introduction; it should be introduced in full as “temperature-humidity index” before using the abbreviation. The phrase “temperature was recorded by the Caotun Weather Station” should specify whether it was recorded automatically or manually. In “TMR was set to 35%,” the units should be specified (e.g., “35% dry matter”). The description of statistical software should be uniform—SAS 9.4 is mentioned, but ensure consistency in formatting and mention specific procedures where applicable
Results
The results are presented logically, but in some cases, the text repeats what is already shown in the tables and figures. Consider summarizing key findings more concisely while referring to the relevant figures for details. The regression analyses provide useful insights, but it would be helpful to include a discussion on potential confounding variables that might have influenced the relationship between ERH and TMR dry matter content. The findings related to high-yield and low-yield groups should be interpreted with caution, as milk production differences may not be solely attributable to TMR consistency.
The phrase “A total of 84 TMR samples were collected” could be more precise—was this across all treatment groups or per group? The sentence “The results showed improved milk yield (p = 0.04)” should specify whether this result was significant based on the defined threshold. In Table 3, the units for “Moisture (%)” should be explicitly stated in the column heading to avoid ambiguity. Some figure legends use “R²” inconsistently; ensure uniform formatting throughout.
Discussion
The discussion does a good job of connecting findings to previous literature, but some claims would benefit from additional references, particularly regarding how ERH fluctuations affect nutrient intake and milk composition stability. The interpretation of TMR adjustments as a management strategy is interesting, but a more detailed explanation of how such adjustments can be practically implemented on commercial dairy farms is needed. Additionally, potential limitations of the study, such as seasonal effects or variations in cow physiology, should be acknowledged.
The phrase “These findings suggest…” should clarify whether this is a direct conclusion from the study or a comparison to prior research. The sentence “Changes in ERH can lead to variations in the DMI of cattle” should specify whether this applies mainly to high-yielding cows or to all production levels equally. In the paragraph discussing FPR values, a reference should be added to support the claim that values below 1.2 indicate acidosis risk.
Conclusion
The conclusion summarizes key findings well, but it would be stronger if it explicitly stated recommendations for dairy farmers regarding TMR adjustments. If any future research directions are suggested, they should be more explicitly linked to the study’s limitations and knowledge gaps.
The phrase “Routine monitoring and precise TMR DM adjustment can enhance production consistency” could be more impactful by specifying the primary benefit (e.g., “can enhance production consistency and reduce fluctuations in milk yield”). The final sentence would be stronger if reworded from “making it a valuable strategy for dairy farms” to “highlighting its potential as a valuable strategy for dairy farm management.”
Author Response

(The authors gave the same response as above.)

Reviewer 3 Report
Comments and Suggestions for Authors
The research content has guiding significance for practical production. The experiments were done very meticulously. But the materials and methods were written very poorly. Many experiments and design were not described clearly. It is the reason why it is difficult for readers to understand their results. Additionally, the description of the results is also very inappropriate. Some results are not displayed in the tables or figures. Please refer to the attachment for details.

Comments on the Quality of English Language
Some sentences have grammar issues.
Author Response

(The authors gave the same response as above.)

Round 2
Reviewer 1 Report
Comments and Suggestions for Authors
I have reviewe the revised version, most of my previous concorns have been addreseed or provided an acceptable explanation. I think it is ready to ask Editors and other reviewers' suggestions. Good luck!
Reviewer 2 Report
Comments and Suggestions for Authors
authors did a great job revising the manuscript, i have no further comments
Reviewer 3 Report
Comments and Suggestions for Authors
After revision, the MS is very improved. But the experimental design is still not explained clearly, especially Experiment 2. I'm confused about what is the "ERH difference"( the horizontal axis in the figure)? Dose that mean ERH difference on the same day or on different days? There are some
designs that I think are meaningless. For example, in the experiment 2, the lactating cows were classified into low-yield and high-yield groups based on a milk production threshold of 30 kg/d. The author did not compare the two groups, so I don't know the significance of this grouping. In addition, the data was not analyzed by analysis of variance in the ms, making the comparisons meaningless. If these issues are resolved, the article will be improved.
Author Response
Clarification of “ERH Difference”:
As stated in Section 2.3.1 Animals and Treatments, the “ERH difference” refers to the variation in environmental relative humidity (ERH) within the same day, not across different days. This is clarified in the following sentence in red from the manuscript:
“…we conducted a correlation analysis between the daily fluctuations in relative humidity…”(p3)
Furthermore, we have revised the captions of Figure 2 and Figure 3 to clearly state:
“ERH difference (%) within a day in high- and low-yield groups for environmental relative
humidity (ERH) difference …”(p8, p9)
Purpose of High- and Low-Yield Grouping in Experiment 2:
The division of lactating cows into high-yield and low-yield groups, using a threshold of 30 kg/day of milk production, was intended to account for differences in nutritional requirements and feeding responses based on production levels. Although a direct statistical comparison between the two groups was not the primary focus of this study, the grouping enabled us to observe how variations in ERH and TMR DM may differentially affect cows with distinct levels of productivity.
This rationale (highlighted in red text) is elaborated in Section 4.2 Effects of Variations in TMR DM on Production Performance (p15) and Section 4.4 Practical Implications for Managing TMR DM Under Varying ERH Conditions (p17), where we emphasize that high-yield cows appeared more sensitive to TMR DM variations under greater ERH differences.
We hope these clarifications better explain the design and intention behind the grouping, as well as the significance of the ERH difference variable used in the study.